# Acquired Hemophilia A: A Permanent Challenge for All Physicians

**DOI:** 10.3390/medicines9030021

**Published:** 2022-03-02

**Authors:** Knut M. Nowak, Alexander Carpinteiro, Cynthia Szalai, Fuat H. Saner

**Affiliations:** 1Department of General, Visceral- and Transplant Surgery, Medical Center University Duisburg-Essen, 45147 Essen, Germany; knut.nowak@uk-essen.de; 2Department of Hematology and Stem Cell Transplantation, Medical Center University Duisburg-Essen, 45147 Essen, Germany; alexander.carpinteiro@uk-essen.de; 3Department of Anesthesiology and Critical Care, Medical Center University Duisburg-Essen, 45147 Essen, Germany; cynthia.szalai@uk-essen.de

**Keywords:** bleeding, acquired hemophilia A (AHA), recombinant activated factor VIIa

## Abstract

Acquired hemophilia A (AHA) is a rare disease with a prevalence in Europe of 1.5 per million. This diagnosis is significantly delayed in about one-third of all cases, leading to deferred treatment. The main signs of AHA are spontaneous bleeding seen in about two-thirds of all patients. AHA can be lethal in 20% of all symptomatic cases. This patient population’s main standard laboratory finding is a prolonged aPTT (activated prothrombin Time) with otherwise normal coagulation results. In addition, antibodies against FVIII (in Bethesda Units) and a quantitative reduction of FVIII activity are necessary to confirm AHA. The therapy of acute bleeding related to AHA is based on the following main principles: Pharmacologic control of the bleeding is of absolute importance. It can be achieved by administering either recombinant activated FVIIa “bypass therapy”; activated prothrombin complex; or Emicizumab, a bispecific monoclonal antibody. Eradication of the FVIII antibodies should be initiated simultaneously. The combination of steroids with cyclophosphamide leads to the highest eradication rates. Causes of AHA may be related to neoplasms, autoimmune diseases, and pregnancy. We report on a patient who underwent four surgical procedures before the diagnosis of AHA was established.

## 1. Case Report

We report on a 68-year-old female patient who developed a retroperitoneal hematoma (12 × 13 cm) after falling at home. The patient underwent multiple surgeries, including abdominal packing and depacking of towels. Despite hemostasis treatment consisting of fresh frozen plasma and Prothrombin-complex (PCC) transfusions, surgical bleeding could not be controlled. The patient was then referred to our facility for further treatment.

At the time of admission, the patient was intubated and ventilated using a bilevel positive airway pressure mode with the following parameters: an inspired Oxygen 50%, positive end-expiratory pressure 12 mbar, inspiratory pressure 30 mbar, and lung compliance of 35 mL/mbar. The patient had an ARDS II according to the Berlin classification (bilateral occupation of both lungs and Horowitz index 148 mmHg) [1].

The standard coagulation parameters showed an isolated aPTT prolongation (72.7 s), while PT/INR and platelets were in the normal range. ROTEM analysis showed a prolonged clotting time in the INTEM assay (see Figure 1). Isolated prolongation of the aPTT; patient age; and the lack of bleeding history together with the acute, spontaneous bleeding propensity suggested the suspicion for AHA. The factor VIII (FVIII) activity was not detectable (<1% activity). A simultaneous assessment of the inhibitor against FVIII was positive. The Bethesda Units (BE) were 15.

Emergency relaparotomy was indicated as the patient’s abdomen was still packed even after bleeding was controlled. Recombinant activated FVII (rFVIIa; NovoSeven^®^) was administered in a dose of 90 µg/kg body weight. Due to the half-life of rFVIIa of 1.5 h, the same dose was readministered every 3 h perioperatively for the next 72 h (cumulative 48 mg per day).

After 72 h, the dose of rFVIIa was halved, and application intervals were increased to 6 h. On the 8th day of admission, infusion of rFVIIa was no longer necessary. Once the diagnosis of AHA was confirmed, eradication therapy was started with prednisone (70 mg/day corresponding to 1 mg/kg body weight) and cyclophosphamide (100 mg/day). After this treatment, there was a significant increase in FVIII from <1% to 9.7% on the 20th day after the start of eradication therapy (last value was measured 4 weeks after admission: 123.4%). As an adjunct treatment, the patient received tranexamic acid (3 × 1 g p.o.) until the FVIII was increased to >5%. When FVIII activity was >20%, a percutaneous tracheostomy was performed to facilitate prolonged ventilation. The patient was successfully weaned from the respirator after 109 days when FVIII activity was >20%. There was a steady improvement in the general condition.

After a total hospital stay of 145 days (118 days at intensive care unit, 27 days in intermediate care unit, and 2321 h of ventilation), the patient recovered adequately from her underlying condition to be discharged for follow-up treatment. The time course of the interventions in the first 6 weeks is shown in Figure 2.

## 2. Discussion

### 2.1. Epidemiology

AHA should always be considered with the occurrence of spontaneous retroperitoneal bleeding and a negative bleeding history, especially in patients between the 6th and 8th decades of life [2]. Due to its rare occurrence, diagnosis and treatment are delayed as in the presented case.

The incidence of AHA is estimated as 1.5 cases/1,000,000 population in Europe [3]. Knoebl et al. [3] evaluated the European Acquired Hemophilia Registry (EACH2), which represented data from 13 European countries, including 117 centers and 501 patients. At the time of diagnosis, the median age was 73.9 years. In 51.9% of patients, AHA was not triggered; in 11.8% of the cases, the diagnosis was associated with either tumor disease or, in 11.6%, with an autoimmune disease. In 33.5% of the cases, bleeding was recognized too late as AHA. Our patient underwent multiple surgeries with a hemorrhagic origin without establishing the diagnosis of AHA.

The German Society for Thrombosis and Hemostasis Research (GTH) established a database in January 2010 to collect all data related to AHA. Besides the significant impact of immunosuppressive treatment for eradicating the inhibitor [4,5], a statement concerning the safety and efficacy of coagulation management could be made [6]. Furthermore, Holstein states that a weekly FVIII assessment would benefit the patient [7].

The cause of AHA is still unknown. However, there are some correlations with malignant tumors (6–18%), autoimmune disease (9–17%), and pregnancy (rate unknown) [8]. Around 68% of the bleeding occurs spontaneously. In most cases, life-threatening bleeding occurs with a mortality of 20% [9].

### 2.2. Diagnostic

Cases of isolated aPTT prolongation that coincide with bleeding should be investigated as AHA. Cases of antiphospholipid syndrome (APLS), which also show a prolonged aPTT, should be ruled out. In contrast to AHA, APLS is characterized by thrombosis rather than bleeding [8]. In addition, specific antibodies are found against different phospholipids (Cardiolipin, Pro-thrombin) and Phospholipid-binding proteins such as beta-2-Glycoprotein I, which increase the risk of thrombosis [10]. Diagnostic accuracy for APLS is based on lupus-anticoagulant (LA) detection. LA antibodies against thrombin and beta-2-glycoprotein compete with the plasma coagulation factors for the binding sites on the phospholipids, which prompts a prolonged aPTT in vitro, although hypercoagulopathy is present [11].

In case of bleeding with an isolated aPTT prolongation, an individual factor determination should be carried out, simultaneously quantifying the inhibitor with the Bethesda assay. The inhibitors are antibodies directed against FVIII. The level of the antibody is given in Bethesda units (BU). The detection of the antibody in connection with the reduced FVIII activity (<5%) confirms the diagnosis of AHA [6]. There is no correlation between inhibitor serum concentration and bleeding. However, lower levels of antibodies, e.g., 5 BE, may cause more severe bleeding than antibodies with higher serum levels (e.g., 15 BE).

### 2.3. Discussion of AHA Treatment

The treatment of AHA can be separated into four stages See Figure 3.

There are several agents available for bleeding control. The agents are stratified into “Bypass” agents and FVIII replacement agents.

#### 2.3.1. Factor VIII Replacement Treatment

Factor VIII agents are effective when the FVIII inhibitor Titer is ≤5 BE [12]. In cases where FVIII inhibitor Titer is ≤5 BE, the FVIII dose should be higher than when compared to inherited Hemophilia A. Higher doses cause effective suppression of the inhibitor, leading to sufficient FVIII activity to provide adequate hemostasis. The use of FVIII was successful in some patients with FVIII inhibitor Titer ≤5% and could be used as a first-line treatment. The recombinant porcine FVIII (rpFVIII; Obizur^®^) represents an alternative to human FVIII preparations. The effectiveness was demonstrated in a study with a limited number of 28 patients [13]. Within 24 h of application of rpFVIII, all 28 patients showed FVIII serum activity > than 100%. In some cases, antibodies against rpFVIII are already present in serum or developed during treatment, making the routine assessment of these antibodies necessary before treatment or especially the response to rpFVIII is blunted. Therefore, the main advantage of rpFVIII agents compared to bypass treatment is the possibility of FVIII activity assessment.

#### 2.3.2. Bypass-Treatment

Both rFVIIa and the activated prothrombin complex aPCC, (FEIBA (factor eight bypassing agent)) are suitable for AHA treatment [8]. The agent aPCC contains the vitamin K-dependent factors II, IX, and X, and the FVII, as activated factors, in contrast to PCC. Here the factors are inactivated and dissolved in heparin.

By administering these agents, propagation and amplification are skipped (see Figure 3), and a thrombin burst occurs directly (hence the term “bypassing agents”), which then activates fibrin formation. In the EACH-2 study [3], most patients were treated with rFVIIa (56.7%), 20.5 % of the patients received aPCC, FVIII agents were used in 18.2% of the cases, and 4.6% received Desmopressin-Vasopressin (DDAVP). The most effective treatment was observed with the “bypass preparations”. Bleeding could successfully be controlled in 91.8% of the cases. There was no statistically significant difference between the two drugs preparations (rFVIIa, 91.8% vs. 93.3% FEI-BA). Replacement with FVIII was less effective than the two above-mentioned agents. Bleeding control was achieved in only 69.9%. (For further details, see Table 1). For this reason, we preferred bypass therapy with rFVIIa (NovoSeven^®^) for our patient. In our case, no other bleeding events were reported after hematoma evacuation and removal of the packed towels, further supporting our choice in hemostasis treatment.

Recently, instead of bypass therapy, monoclonal antibodies, such as Emicizumab (Hemlibra^®^) have been used [14]. Emiczumab is a recombinant, bispecific, monoclonal antibody that mimics the function of activated FVIII. Emicizumab binds with one side to the FIXa, the other side to the zymogen FX, and then activates the conversion of FX to F Xa. There is no structural relationship to FVIII, which prevents the formation of antibodies [15,16]. The drug has two decisive advantages. First, the possibility of subcutaneous application leads to extensive and independent use by the patient, which does not require medical consultation. The second very decisive advantage is the long half-life of 30 days, which makes the agent very attractive as a prophylactic preparation for patients with inherited Hemophilia A [17]. Thomas et al. conducted a scoping review on the use of emicizumab in AHA [18]. The authors reported on 33 AHA patients, from 12 studies. The patients presented with a wide range of bleeding symptoms: hematoma formation, mucocutaneous bleeds, and surgical site bleeds. Of these 33 patients, 26 received various methods of hemostasis treatment (rFVIIa, aPCC or rpFVIII). All patients achieved a clinical response to emicizumab with no further spontaneous bleeding.

Further supporting hemostasis treatment is based on the increased release of FVIII and von Willebrand factor from the subendothelium. In some case reports, the use of TXA in combination with aPCC or rFVIIa was effective in increasing the clot strength [19,20]. From that point of view, TXA can be considered as a complementary therapy.

Treatment with aPCC or rFVIIa carries a significant risk of thromboembolic events. In the literature, the rate varies between 0% and 55%. However, studies that recruited at least 100 patients report an incidence between 2.3% and 2.9% [21,22,23]. Besides a general propensity for thrombosis, factors such as immobility, presence of malignancy, previous thromboembolism, artificial valves, atrial fibrillation, and vascular stents may also contribute to a higher incidence of thrombosis [9].

### 2.4. Inhibitor Eradication

Although some older studies report spontaneous eradication of the inhibitor after several months [24], it is currently recommended that bypass therapy should be commenced simultaneously with eradication therapy [8]. The following three agents, corticosteroids, cyclophosphamide, and rituximab, work effectively for eradication (see Table 2). The median time of eradication was reported as 5 weeks. [8]. Patients who had an FVIII activity <1% at the beginning of the therapy required a significantly longer treatment time for successfully eradicating the inhibitor than patients with FVIII activity of>1%.

In the EACH-2 study, 142 patients received steroids only, 83 patients received steroids and cyclophosphamide simultaneously, and 51 patients received steroids and rituximab simultaneously [25]. The highest remission rate with the shortest time to achieve remission was achieved using steroid plus cyclophosphamide combination (80%, median 40 days). The second-best results were achieved with the steroid + rituximab combination (61%, median 64 days). Steroid therapy alone resulted in remission in only 58% of the cases, with a median time to remission of 32 days. Rituximab therapy alone was rare (12 patients) and had the lowest remissions at 42%. A meta-analysis of 20 studies with 249 patients also concluded that the combination treatment of steroid + cyclophosphamide is superior to steroids alone [26]. Therefore, based on the available data, the authors favor treatment with steroids in combination with cyclophosphamide as first-line therapy.

Due to the mechanism of action of rituximab and the fact that B cells play an essential role in maintaining the immune response, rituximab therapy leads to an increased risk of developing infections. Patients undergoing rituximab therapy should receive a pneumococcal vaccine prior to therapy because of the possibility of iatrogenic B-cell depletion [27].

Cyclophosphamide leads to single and double strand breaks in rapidly dividing cells (bone marrow, mucous membranes), resulting in side effects such as leukopenia, thrombopenia, nausea, vomiting and hemorrhagic cystitis. Simultaneous administration of Mesna (mercaptoethanesulfonate sodium) is intended to prevent this side effect [28]. Since the dose of Cyclophosphoramide in our patient was significantly lower than in oncological patients, simultaneous protective therapy with Mesna was avoided. Hemorrhagic cystitis did not occur in our patient during the cyclophosphamide therapy.

### 2.5. Do Patients with AHA and Bleeding Requires Thrombosis Prophylaxis?

Due to the rarity of the disease, a general statement cannot be made. Data on thromboembolism appear to be inconsistent. In cohorts evaluating at least 100 patients, the incidence seems to be a maximum of 3%. Experts recommend thrombosis prophylaxis with low molecular weight heparin, e.g., enoxaparin at 1 mg/kg, as soon as the FVIII activity is ≥50%. However, there are patient-predisposing factors for thrombosis, such as a history of thrombosis, pregnancy, or the presence of malignant diseases. In these patient groups, thrombosis prophylaxis at an earlier point in time may be considered. In our patient, thrombosis prophylaxis had already been initiated 1 week before the FVIII activity had reached 10% without any signs of bleeding.

## 3. Conclusion for Daily Practice

In the case of unexplainable atraumatic or recurrent bleeding after surgical therapy, a diagnosis of AHA should always be considered.
An isolated prolonged PTT value or prolonged clotting time in INTEM assay, with normal values in standard laboratory parameters or ROTEM (beside CT in INTEM), is suspicious for AHA. FVIII activity and possible inhibitors should be determined in these cases.The standard of care is based on controlling the bleeding with mainly “bypass agents” and antibody eradication therapy.Surgical therapy should only be considered for avoiding nerve compression, circulatory disorders, or other such emergency indications.

## Figures and Tables

**Figure 1 medicines-09-00021-f001:**
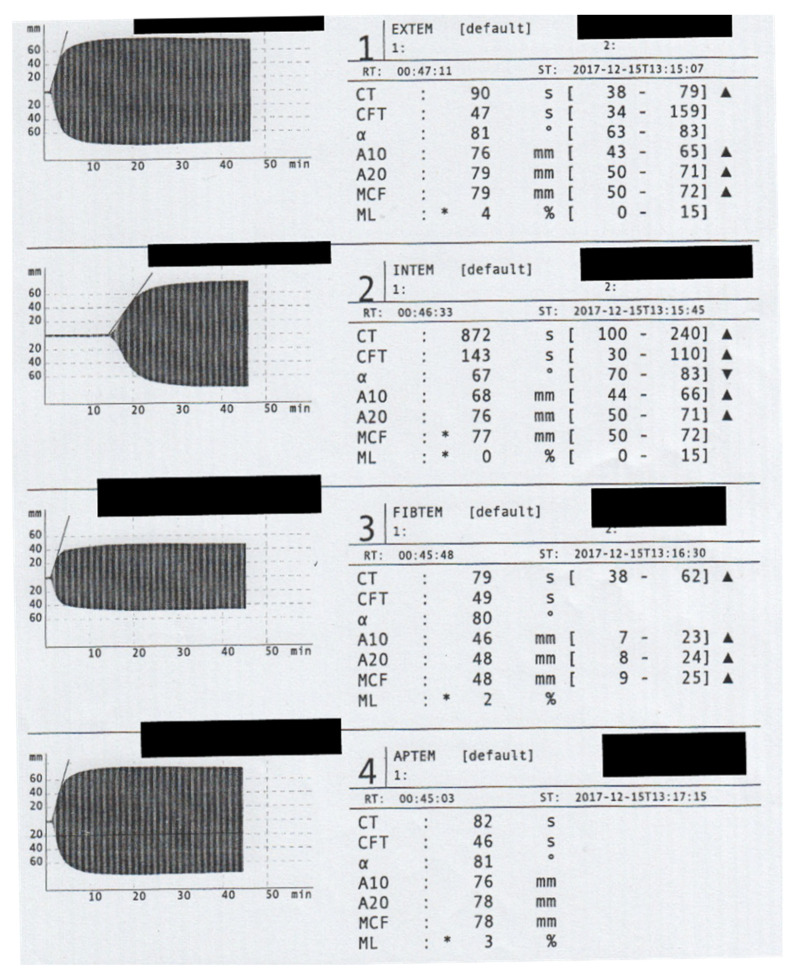
Rotem assessment on Admission. The Rotem analyses showed a significant prolongation of the clotting time (CT) in the INTEM channel. To prevent Heparin administration, we did not conduct a HEPTEM assay. The maximum clot firmness (MCF) in the EXTEM assay (79 mm) is mainly caused by the extremely high fibrin polymerization recorded in FIBTEM channel (MCF = 48 mm).

**Figure 2 medicines-09-00021-f002:**
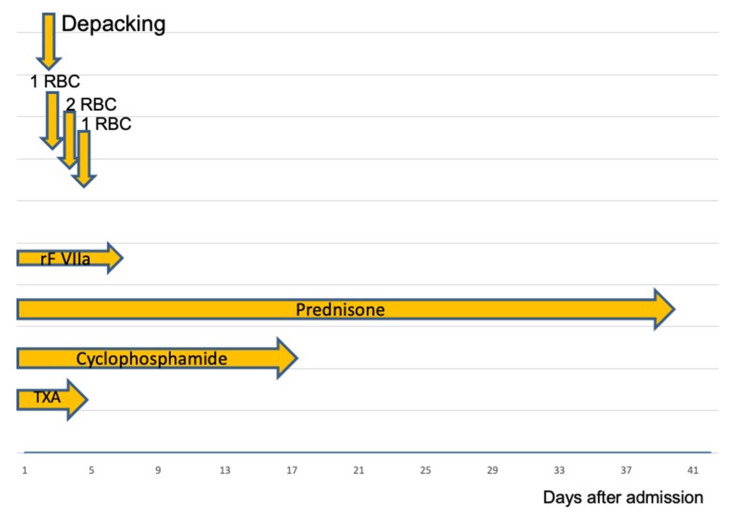
RBC: packed Red Blood Cells; rFVIIa: recombinant Factor VIIa, TXA: Tranexamic acid. After successful coagulation management, the bleeding was stopped. On the second admission day, abdominal packs were removed. During the initial 4 days, the patient received a total of four units of RBCs. Following surgery, the dose of rFVIIa was reduced and then stopped after 3 days.

**Figure 3 medicines-09-00021-f003:**
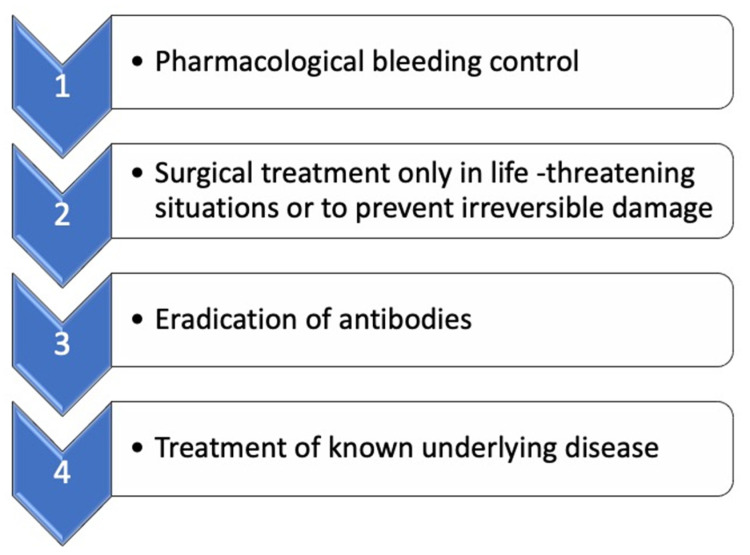
Acute treatment of bleeding. Modified according to Kruse-Jarres et al. [8].

**Table 1 medicines-09-00021-t001:** Drugs used to treat AHA-associated bleeding. Modified according to Kruse-Jarres et al. [8].

Agent	Advantage	Disadvantage
rpFVIII (50–100 IU/kg); FVIII activity assessment every 3 h. Assessment of Ab against rpFVIII => If evident, increase the dose up to 200 IU/kg	Serum levels of FVIII could be monitored	Possible cross-reaction with FVIII-inhibitors => less effectivity
aPCC 50–100 IU/kg every 8–12 h	Always effective even at high levels of AHA inhibitors >10 BE	No possible monitoring, thromboembolic events are possible
rFVIIa 70–90 µg/kg every 3 h until bleeding is stopped; then maintain the dose and extend the intervals	Always effective even at high levels of AHA inhibitors >10 BE	No possible monitoring, thromboembolic events are possible short half-life: 2–3 h

rpFVIII: recombinant porcine Factor VIII. aPCC: activated Prothrombincomplex. rFVIIa: recombinant activated Factor VII.

**Table 2 medicines-09-00021-t002:** Drugs for eradication; modified according to Kruse-Jarres [8].

Recommended Treatment	Dose	Comment
Prednisone	1 mg/kg p.o. 4 weeks	Ineffective for patients with FVIII <1% and inhibitor titer >20 BE Side effects: hyperglycemia, infections, steroid psychosis
Prednisone + Cyclophosphamide	Prednisone 1 mg/kg Cyclophosphamide: 1–2 mg/kg po daily or 5 mg/kg iv for 4 weeks	May shorten the course of the disease More side effects Highest cure rate Bone marrow toxic (leukopenia/thrombopenia)
Prednisone + Rituximab	Prednisone 1 mg/kg Rituximab 375 mg/m^2^ iv/week for 4 weeks	Rituximab is only recommended if the above regimes have failed; Pneumococcal vaccination recommended

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
