# Peer review of "Acquired Hemophilia A: A Permanent Challenge for All Physicians"

_medicines, 2022, doi:10.3390/medicines9030021_

Round 1

Reviewer 1 Report

This paper, more than a case report, is a review article. As a review article, it does not provide any advance in knowledge about acquired haemophilia. Furthermore, this review is independent of the case report, does not take it into account and does not attempt to explain the management of this case.

As a case report on acquired hemophilia A, it has some flaws, gaps and inconsistencies. Some unanswered questions: did the patient have previous bleeding in addition to the traumatic retroperitoneal hematoma? Did the patient have coagulation parameters evaluated before the surgeries? Why did the patient have multiple surgeries before being sent to study? Why the surgeries?  What was the evolution of the hematoma? Why did the patient undergo fresh frozen plasma and PCC transfusions (based on clinical bleeding or impaired laboratory coagulation parameters)? What assessment was done to exclude possible causes of acquired hemophilia?

Author Response

Reviewer 1

This paper, more than a case report, is a review article. As a review article, it does not provide any advance in knowledge about acquired haemophilia. Furthermore, this review is independent of the case report, does not take it into account and does not attempt to explain the management of this case.

As a case report on acquired hemophilia A, it has some flaws, gaps and inconsistencies. Some unanswered questions: did the patient have previous bleeding in addition to the traumatic retroperitoneal hematoma? Did the patient have coagulation parameters evaluated before the surgeries? Why did the patient have multiple surgeries before being sent to study? Why the surgeries?  What was the evolution of the hematoma? Why did the patient undergo fresh frozen plasma and PCC transfusions (based on clinical bleeding or impaired laboratory coagulation parameters)? What assessment was done to exclude possible causes of acquired hemophilia?

Answer: Respected Reviewer 1.

Many thanks for your comment. All your questions are addressed to the referring governmental hospital. They reported that the lady was fallen at home and called an ambulance. At this point the referring hospital assumed that hematoma was related to the collapse. Almost all operations were done in the referring hospital. They did not made a specific coagulation assessment although the regular lab with isolated aPTT prolongation with normal PT and fibrinogen and platelets should already suspicious enough for AHA. The patient was referred to our hospital with the comment that they are not able to control the bleeding. When the patient was admitted to our hospital we performed first, beside regular hemostasis assessment, F VIII and assessment of the inhibitor. When diagnosis of AHA was confirmed, rFVIIa was replaced and then the towels from retroperitoneum was removed. FFP and PCC was given in the referring hospital not in our hospital.

Concern your comment: “does not take it into account and does not attempt to explain the management of this case.” The first three pages are completely dedicated for the case, how we manage the patient, including a timeline figure, when and how we manage the patient.

Reviewer 2 Report

This is a case report describing the delayed diagnosis of AHA in a patient that underwent multiple operations dure to a traumatic retroperitoneal bleed. After admission to the center or the authors, she was properly diagnosed and treated according to current international guidelines. Even if not novel, the case is interessting and pinpoints the need for early diagnosis. The authors then discuss epidemiology, pathophysiology, diagnostics and treatment in some detail. None of this information is new and could be shortened into a brief "introduction" paragraph. I would suggest to remove the "pathophysiology part" based on a 20 year old paper, of which most of the readers should be familiar. Moreover the manuscript is burdened a substatnial amount of typos (legends to fig 1 and 3,

P4. l101, 106, 107, 115, P5

P5. l154, 

P6. 169, 189,190

P8. L255

Moreover, the authors describe serum levels of coagulationfactors and how the coagulationprocess occurs in serum. To my knowledge serum is defined as the fluid that is removed after coagulation has occured, thus, depleted of both cells and coagulation factors.

In the discussion part Emicizumab is mentioned as a possible treatment, though, so far, only few case reports describes its use in patients with aqcuired haemophilia. One important aspect of this product is that steady state levels of the antibody is not reached until 2-4 weeks, which hamper the treatment of acute bleeds.

Some minor comments:

-coagulations factors are most often abbrieviated as FVIII, FVII and so on, with out space between F and roman number. If space is used it should be used consitently

-Why put citation marks on tissue factor pathway inhibitor? (P4, r.118).

Author Response

REVIEWER 2

This is a case report describing the delayed diagnosis of AHA in a patient that underwent multiple operations dure to a traumatic retroperitoneal bleed. After admission to the center or the authors, she was properly diagnosed and treated according to current international guidelines. Even if not novel, the case is interessting and pinpoints the need for early diagnosis. The authors then discuss epidemiology, pathophysiology, diagnostics and treatment in some detail. None of this information is new and could be shortened into a brief "introduction" paragraph. I would suggest to remove the "pathophysiology part" based on a 20 year old paper, of which most of the readers should be familiar. Moreover the manuscript is burdened a substatnial amount of typos (legends to fig 1 and 3,

Reply: Dear reviewer 2 we appreciate your comments and have deleted the paragraph with pathophysiology, including the graph.

The errors have been  corrected

P4. l101, 106, 107, 115, P5

P5. l154, 

P6. 169, 189,190

P8. L255

Moreover, the authors describe serum levels of coagulation factors and how the coagulation process occurs in serum. To my knowledge serum is defined as the fluid that is removed after coagulation has occured, thus, depleted of both cells and coagulation factors.

We apologize for this issue, the mistake has been corrected.

In the discussion part Emicizumab is mentioned as a possible treatment, though, so far, only few case reports describes its use in patients with aqcuired haemophilia. One important aspect of this product is that steady state levels of the antibody is not reached until 2-4 weeks, which hamper the treatment of acute bleeds.

We are very happy for this comment. We added following paragraph to the text:

Thomas et al. conducted a scoping review on the use of emicizumab in AHA (18). The authors reported on 33 AHA patient, resulting from 12 studies. The patients presented with a wide range of bleeding symptom: hematoma formation, mucocutaneous bleeds, and surgical site bleeds. Of these 33 patients 26 received various methods of hemostasis treatment (rF VIIa, aPCC or rp F VIII).  All patients achieved a clinical response to emicizumab with no further spontaneous bleeding.

Some minor comments:

-coagulations factors are most often abbrieviated as FVIII, FVII and so on, with out space between F and roman number. If space is used it should be used consistently

Now between factor and roman numeral all spaces have been  consistently removed.

-Why put citation marks on tissue factor pathway inhibitor? (P4, r.118).

This paragraph was deleted.

Round 2

Reviewer 1 Report

This paper lacks originality and offers no advance in knowledge of acquired haemophilia.

The authors were unable to answer the questions raised.

Author Response

/